# Social Class and Private-Sphere Green Behavior in China: The Mediating Effects of Perceived Status and Environmental Concern

**DOI:** 10.3390/ijerph20054329

**Published:** 2023-02-28

**Authors:** Long Niu, Chuntian Lu, Lijuan Fan

**Affiliations:** Department of Sociology, School of Humanities and Social Sciences, Xi’an Jiaotong University, Xi’an 710049, China

**Keywords:** social class, objective social class, perceived social status, private-sphere green behavior, environmental concern

## Abstract

Green behavior is traditionally considered as an effective way to ameliorate environmental degradation and requires an individual to make sacrifices of their social resources. However, few studies have focused on its status signaling. In this study, we draw on the theory of social class and the concept of status signaling theory to empirically investigate the effect of objective social class and perceived social status on private-sphere green behavior in China. Using national comprehensive survey data from China General Survey Data (CGSS) in 2021 subjected to ordinary least-square regression model and step regression models, we present the following results: (1) higher-class individuals, both objectively and subjectively, tend to engage in more private-sphere green behavior than their lower-class counterparts; (2) the effect of objective social class on private-sphere green behavior is mediated by individual’s perceived social status in the class hierarchy; (3) environmental concern significantly correlates with private-sphere green behavior, and it also mediates the effect between objective social class and private-sphere green behavior. The present research provides insights into how social class and its psychological manifestations (i.e., perceptions of status) correlate with private-green behavior in China. Our results suggest that more social context factors should be considered when identifying the factors promoting pro-environmental behavior in China.

## 1. Introduction

Since the industrial revolution, human actions have changed environment dramatically, making the planet we live on a place facing unprecedented crisis. Under the threat of desertification, global warming, deforestation and constant extinction of species worldwide, these issues are viewed by social and natural sciences as some of the most pressing problems facing modernity. This is not only because of the deleterious effect of environmental degradation, but also because of the necessity to mitigate these effects, that has aroused global questions focused on identifying those factors that can influence behavior to ameliorate environmental problems. One of the most effective ways is to advocate individuals to conduct green behavior in the private sphere [1,2]. Private-sphere green behaviors, such as green consumption [3,4,5,6], adopting eco-friendly lifestyles and behaviors, purchasing vegetables without chemicals, taking public transportation, conducting more recycling to reduce negative environmental impacts, retrofitting existing buildings into green buildings to reduce energy consumption [7,8] or promoting the breadth of environmental concern [9]. It is expected that shifting an individual’s preference toward green behavior and arousing general environmental concern could reap long-term environmental benefits [10,11].

Previous studies have proposed macro-level and micro-level determinants of green behavior that should, in practice, lead to a greater good for nature. These studies have made substantial headway in subjecting these determinants to empirical testing. Macro-level studies often use country-level variables, including geographic factors such as developed countries vs. undeveloped countries [12], western countries vs. African countries [13], etc. Other macro-level variables, such as GDP, income inequality, inflation, unemployment, are repeatedly considered in order to examine environmental attitudes, concerns and behaviors [14,15,16]. Micro-level studies have considered the roles of individuals’ sociodemographic, knowledge, values and attitudes, as well as more socially oriented influences on green behaviors [9,17,18,19]. As social stratification and wealth inequality become more severe in many societies, it is critical to better understand how social class correlates with environmental behavior [20,21,22,23,24]. This knowledge would shift our understandings of what green behavior is “about” to recognizing those who would “participate” to protect environment. What’s more, some researchers argue that green behavior is not merely a reflection of an environmental ethos, but that it can also act as a vehicle for signaling class status, and that this green behavior sometimes appeals to people for this reason [10]. However, prior research has often viewed social class as a background variable or as a basis for segmentation [25,26], and studies about social class often treat objective or subjective social class separately. In this study, we argue that objective and subjective social class might influence an individual’s behavior in a different way because subjective social class is a psychological construct. The differences in objective social class and subjective social class highlight the necessity to distinguish the impacts of such disparities on green behavior [27]. Thus, how social class influences green behavior requires further investigation. With this in mind, this study asks the following research questions: How does social class, both objective and subjective, influence private-sphere green behavior in China? Are the relationships between social class and private-sphere green behavior mediated by subjective social class and environmental concern?

To address these questions, we draw on the theory of social class and status signaling and propose the green conspicuousness framework to examine the influence of objective/subjective social class on green behavior. The present study aims to take a nuanced look into the effects of different social class factors on green behavior. We expect that those of a higher social class would conduct more green behavior in private sphere. Conducting green behavior requires individual sacrifice (such as time, money, etc.). However, being able to make such sacrifices reveals one’s ability (status signaling). Thus, we posit that an individual’s perception of higher status rank might elicit pro-environmental behaviors in order to demonstrate their social status. We examine this framework by analyzing the mediating effect of perceived status rank between objective social class and private-sphere green behavior.

The paper suffers from three research limitations. First, most of the research concerning social class and green behavior are focused on either western developed countries or African undeveloped countries; studies in developing countries (i.e., China) are quite scarce. As the biggest developing country, China has the largest population and is the one of the top global emitters of greenhouse gases in the world. Promoting green behavior among Chinese individuals would be a significant boost in protecting the environment. Second, prior research has often viewed social class as a background variable or as a basis for segmentation. Few studies have explored the correlations between objective/subjective social class and green behaviors simultaneously, especially considering that subjective social class has psychological manifestations, and might influence individual’s behavior in different ways [28,29]. In general, research on how social class influences private-sphere green behavior is still in its infancy [30]. Thus, we propose multi-level class factors, measured objectively and subjectively, to investigate the separating effect of social class on green behavior. Furthermore, we regard objective social class as a predictor and subjective social class as a mediator to further examine how objective social class affects green behavior through subjective social class. Finally, as environmental problems become a global issue, some researchers argue that environmental concern has widespread manifestations in different countries, and do not necessarily conform to postmaterialist values or affluence outcomes [31]. Thus, we also examine the mediating effect of environmental concern between objective social class and private-sphere green behavior as well. The present study provides a nuanced look into how social class, both objective and subjective, influences private-sphere green behavior in China. By analyzing the mediating effect of perceived status between objective social class and green behaviors, this study takes a closer look at the psychological manifestation of objective social class in influencing individuals’ green behavior.

## 2. Literature Review and Hypothesis Development

### 2.1. Social Class and Green Behavior

Green behavior is prosocial, as individuals intend for others to be the beneficiaries of their pro-environmental behaviors [32] and it sometimes requires individual self-sacrifice and willingness to forgo personal interest for the benefit of the environment [33,34]. Consider green products on the market. They are often perceived to be of lower quality and cost more money, while being prosocial and having higher ethical standards [35]. Thus, conducting green behavior may cause certain burdens, such as paying higher prices for eco-friendly products. The ability to carry these burdens is part of the decision to perform green behaviors [36]. In this sense, people who possess more social resources tend to perform more green behaviors to protect the environment as they do not have to overcome the same basic materialist needs as lower-class individuals do.

Social class is one of the most salient attributes of a society. It is multifaceted in nature [37], reflecting individual differences in power [38], wealth [39], financial status [40], cultural capital [41] and social resources [42]. It is not only defined by an individual’s objective access to material resources (e.g., income, education, and occupational prestige) but also corresponds to the subjective perceptions of status rank vis-a’-vis others [43,44]. Thus, social class can be defined in objective and subjective ways [27]. As a ubiquitous force shaping one’s identity with psychological and cultural underpinnings [43,45], social class permeates many aspects of our lives. On the one hand, social class has profound impacts on consumer cognition, emotion and behavior [46], including their thinking style [47], empathic accuracy [48], moral judgement [39], product perception and assessment [40] and ethical or prosocial behavior. On the other hand, upper-class individuals prioritize their need to be individuated, more distinct and privileged in social rank than their lower-class counterparts, who possess constrained resources and have greater daily struggles [49].

Based on the above analysis of social class and green behavior, we propose our conceptual framework to determine the direct relationship between objective social class and private-sphere green behavior and to identify the mediating mechanisms (i.e., perceived status and environmental concern). Figure 1 depicts our conceptual framework.

### 2.2. Objective Social Class and Green Behavior

Though not directly specified, previous research about the determinants of green behavior and environmental concern reveals the potential effect of affluence, a factor deemed important by most upper-class individuals. The dominant research comes from Inglehart’s post-materialism values. In his groundbreaking study, Inglehart et al. (2014) argue that more affluent societies have higher environmental concern and, within countries, wealthier individuals care more about the environment [50]. Another thesis that suggests affluence as a source of environmental concern is the prosperity or affluence hypothesis. According to these scholars, environmental quality is both a public and normal good, and individuals demand higher environmental quality as their income rises. For example, analyzing data for 21 countries from the International Social Survey Program (ISSP), Diekmann and Franzen (1999) find that nine of the eleven environmental concern variables they examined were positively and significantly correlated with GNP per capita [51]. Additionally, Franzen (2003) used data from ISSP for 1993 and 2000 to examine environmental concern across countries and observed that wealth significantly influences environmental concern [52]. More recently, Franzen and Meyer (2009) also found support for the prosperity or affluence hypothesis [15]. Among economist, the environmental Kuznets curve (EKC) hypothesis is often used to explain environmental concern [53]. According to his hypothesis, there is an inverted U-shaped relationship between income and environmental degradation. In other words, as economies grow, environmental pollution rises initially. However, once a certain turning point in income is reached and exceeded, people begin to demand higher environmental quality.

This analysis lends a convincing explanation about how affluence correlates with green behavior. The conclusions are consistent, arguing that the growing affluence brings higher environmental concern. Thus, when facing the high cost of bearing green behavior, higher-class individuals tend to show a greater ability to afford the cost and conduct more green behaviors than their lower-class counterparts. Based on the analysis above, we hypothesize that:

**Hypothesis** **1.**
*There is a positive relationship between objective social class and private-sphere green behavior.*


### 2.3. “Green Conspicuousness”: The Influence of Perceived Social Status on Green Behavior

Bourdieu’s conceptions of taste and of status signal provide useful theoretical frameworks in understanding of status and social class in daily lives. In his Distinction Theory (1984), he argues against the abstract rational chooser of economic models and gives us, instead, a more circumscribed actor: one who possesses historically and structurally derived resources, abilities and expectations. The habitus of any given individual is constituted by the volume and structure of his/her economic capital (wealth), producing both practices and “the capability to differentiate and appreciate these practices and products,” or taste [54]. In this way, these individuals relate to each other based on their economic capital, their educational level, their family background and their social trajectory. In this way. the habitus is “a structuring structure” [54].

Previous consumer behavior research has indicated that social class consciousness will influence an individual’s preferences for goods or services [55]. Individuals with a strong desire for higher social class rank are more likely to buy goods or services that allow them to demonstrate their wealth-related aptitude [56]. Cultural sociological literature on environmental topics reveals that high-status consumption practices now seem to embody a new “ecological orientation of high-status tastes” [57]. Griskevicius et al. (2010) conclude that “a key component of harnessing the power of status motives to benefit social welfare necessitates that the prosocial acts be visible to others, whereby such acts can clearly influence the well-doer’s reputation”. In an experimental study, he found that “activating status motives” led individuals to choose green products over more luxurious nongreen products [10]. Other studies reveal that social class and pro-environmental behaviors related to the demonstration of higher-class status [58]. Sexton and Sexton identified a statistically significant “conspicuous conservation effect” in vehicle purchases and a willingness to pay for a “green” signal, that they attribute to a status-motivated desire to display “austerity rather than ostentation.” [34]. Similarly, Berger found that individuals seeking to reaffirm their class status prefer socially visible green consumption that elevates their social standing in the eyes of others [59]. Based on the analysis above, we hypothesize that:

**Hypothesis** **2.**
*There is a positive relationship between perceived social status and private-sphere green behavior.*


### 2.4. The Role of Perceived Social Status

Perceived social status is captured by individuals’ subjective perceptions of rank vis- a’-vis others [60]. If objective social class represents the amount of actual social resources, perceived social status or subjective social class is psychologically constructed based on those objective resources. However, significant resources do not necessarily lead to higher perceived status consciousness, and vice versa [61]. There seem to be certain disparities between objective and subjective social class [27]. Prior research tends to operationalize social class as dichotomies, such as lower vs. upper class, and lower vs. higher SES [48], emphasizing a clear distinction and comparison between low and high social class groups [43]. Nonetheless, this heuristic simplification overlooks nuanced differences about those factors constructed by the social class. In this paper, we acknowledge the disparities between these two types of social class, and regard perceived social status as the outcome of objective social class and the determinant of private-sphere green behavior. In other words, we treat perceive status as a mediator, investigating the mediation effect between objective social class and private-sphere green behavior.

Recent research suggests that the effects of social class on behavior are complex, rather than simple [37]. In particular, the relationship between social class and prosocial behavior is more equivocal and nuanced than earlier findings suggested. These studies see the necessity to explore the psychological manifestation of social class in green behavior. In fact, some researchers argue that the effects of social class are best captured under the umbrella of a subjective rank-based measure [60]. For example, previous research has explored the correlations between perceived socioeconomic status and individuals’ willingness to undergo income sacrifice to protect the environment. These studies empirically found that the higher an individual perceives their socioeconomic status, the greater the willingness to sacrifice for the environment [62]. Based on the analysis above, we hypothesize that:

**Hypothesis** **3.**
*Perceived social status mediates the effect between objective social class and private-sphere green behavior.*


### 2.5. Environmental Concern

Environmental concern refers, as Dunlap and Jones point out, “to the degree to which people are aware of problems regarding the environment and support efforts to solve them and/or indicate a willingness to contribute personally to their solution” [63]. Environmental concern is thought to be the prerequisite of green behavior, with individuals who have a higher concern about environment tending to be greener than individuals with a lower concern. Miller et al. (2022) used data from a large international survey in 11 countries and showed that environmental attitudes are a strong predictor of pro-environmental behaviors [13].

Theoretically, various forms of prosocial behavior are associated with different social-cognitive demands and motivational antecedents [64]. Although social class is derived from economic inequality, the attitude that individuals view nature within the social hierarchy varies. Prior research shows that individuals’ concern about the environmental varies by wealth, culture and prestige, and even within an organization or society [6]. Since the late 1980s, research on consumers has, in general, increasingly focused on the creation of “psycholographic” profiles of consumer segments based on responses to questions about self-conception of and preferences about “values and lifestyle surveys” (VALS), as a supplement to or even replacement for basic socio-demographic profiles [65]. In a review of this literature, Diamantopoulos et al. (2003) conclude that, while socio-demographics do appear to be related to green consumption, they are of limited explanatory value in isolation, lending support to a more VALS-based market segmentation approach [66]. Building on prior research suggesting that perceived concern about the environment would affect the relationship between social class and pro-sociality [67], we propose that environmental concern would mediate the effect of social class on green behavior. Thus, we hypothesize that:

**Hypothesis** **4.**
*There is positive relationship between environmental concern and private-sphere green behavior.*


**Hypothesis** **5.**
*Environmental concern mediates the relationship between objective social class and private-sphere green behavior.*


## 3. Method

### 3.1. Data

To empirically test whether social class influences green behavior, this study used the Chinese General Social Survey (CGSS) from 2021. CGSS is a national, synthetical and consecutive academic program in China that gathers information of residents regarding family structure, education, employment, health, social attitudes, etc., to observe the characteristics and trends of Chinese society and discuss social issues with theoretical and practical significance. The CGSS employed a multi-stage stratified sampling procedure, with counties serving as primary sampling units, urban communities and rural villages as secondary sampling units and households randomly selected using a mapping sampling method [68]. The representative samples and broad range of topics have been acknowledged and widely used by the academic community. We chose the more recent dataset, from 2021. It comprises measurements of respondents’ environmental awareness, attitudes and behaviors, providing a glimpse of Chinese environmental concern and actions. In the 2021 survey, about 33.6 percent of respondents (N = 8148) were randomly assigned to answer the questions in the environmental module (N = 2741). The final sample was 2703 after deleting the core missing values.

### 3.2. Measures

#### 3.2.1. Dependent Variable

Private-sphere green behavior. There are many classical structures formed when measuring green behaviors, such as five-division classification and the four-category scheme. According to Sia et al. (1985), green behaviors can be divided into five categories: ecological management behaviors, consumer behaviors, persuasive actions, political behaviors and legal behaviors [69]. Stern classified green behaviors into four categories: environmental activism, non-activist behaviors in the public sector, private-sector environmentalism and other ecologically relevant behaviors [2]. Larson et al. (2015) explored the multi-dimensional structure of green behaviors and categorized them as follows: conservation lifestyle behaviors (e.g., household actions in the private sphere), social environmentalism, environmental citizenship and land stewardship [1]. In this paper, we define green behavior in a simple categorization in the private sphere. Private-sphere green behaviors are measured by three items in CGSS (2021): “I am always categorizing the household waste.”; “I reuse household products.”; “I would like to communicate with other residents about how to categorize waste.”. All the questions are scaled 1 to 5: 1 for strongly unwilling and 5 for highly willing. In our analysis, a higher score demonstrates a higher frequency of green behavior.

#### 3.2.2. Independent Variables

*Objective social class.* In this paper, we indexed objective social class with wealth, education and prestige of occupation. Wealth fundamentally reflects the differences in the objective resources that give rise to class divisions, and the amount of wealth may act as a proxy for a more broad social position [70]. Wealth was measured by asking the respondents “What is your total household income in past year?”. We take the logarithm of the household income to obtain a normal distribution. A higher score indicates a higher objective social class.

*Perceived social status*. The CGSS measured respondents’ subjective social class using the McArthur scale of subjective SES, a rank-based measure used to assess personal placement compared to others within a resource-based social hierarchy [71]. It reflects the convergence of objective resources and subjective social ranks [60]. Respondents saw an image of a 10-rung ladder representing where people stood in their local community in terms of income, education and job status. They were asked to select a rung number to represent their position relative to others (1 = bottom rung, 10 = top rung). The higher value demonstrates higher subjective social class rank in the society.

*Environmental concern.* The New Environmental Paradigm (NEP) scale was used to measure environmental concern [63]. The NEP is designed to evaluate five aspects of an individual’s environmental worldview: the realization of limits to growth, anti-anthropocentrism, belief in the fragility of the balance of nature, rejection of human exceptionalism and belief in future eco-crisis. The respondents were presented fifteen statements and asked to select a value from a 5-point Likert-type scale to indicate the extent to which they agree with each statement (from strongly agree to strongly disagree). Although some studies indicated that the NEP has multiple dimensions, e.g., balance of nature, limits to growth and human domination of nature [63,72], high internal consistency (Cronbach’s alpha > 0.7) is typically considered justification for aggregating all 15 statements in a scale that can range from 8 to 40 (a higher total score indicates a more pro-environmental value) [63]. The NEP scale measured in this dataset had a mean value of 28.36 and had moderately high internal consistency (Cronbach’s alpha is 0.75).

#### 3.2.3. Control Variables

We controlled for sociodemographic variables in the present study, comprising gender, age, marriage and religious affiliation [62,73,74]. Gender was measured using a dummy variable (0 = female, 1 = male). Age was a continuous variable with each respondent providing his or her birth year; we computed their age by 2021 minus their birth year. In regards to the complexity of marriage, marriage was classed as “0 = not in marriage” and “1= in marriage” for simplicity. We categorized religious affiliation into “0 = not religious” and “1 = religious” when respondents answered whether they belonged to a specific religion.

### 3.3. Analysis Strategy

Prior research has seldom combined the effects of objective and subjective social class on green behavior into one framework. In this study, we tested the effects of both objective and subjective social class using the ordinary least-square regression model. In this study, we first examined the impact of objective social class, perceived social status and environmental concern on private-sphere green behavior.
$$Y_{i}=\alpha +\beta_{i}X_{i}+\gamma_{i}CONTROL+\varepsilon$$

The equation above denotes the standard OLS regression model, whereby $Y_{i}$ indicates the effect of objective social class, perceived social status and environmental concern on private-sphere green behavior. $\beta_{i}$ denotes the effect size of $X_{i}$. $\gamma_{i}$ denotes the effect size of control variables and ε is estimation error.

Then, we conducted stepwise regression to explore the mediating effects of perceived social status and environmental concern between objective social class and private-sphere green behavior [75]. Stepwise regression models are used to examine the relationship between an independent variable (X), mediator (M) and dependent variable (Y). It has three steps: first, run a standard regression model to examine whether the coefficient c is in significance. If c is significant, then run the regression of Equations (2) and (3) to examine whether coefficient a and b are in significance. Finally, check the significance of coefficient $c^{\prime }$. If $c^{\prime }$ is significant, this indicates that the mediator partially mediates the relationship between (X) and (Y). Elsewhere, (M) is a full mediator if the coefficient $c^{\prime }$ is insignificant.
(1)$$Y=cX+e_{1}$$
(2)$$M=aX+e_{2}\;$$
(3)$$Y=c^{\prime }X+bM+e_{3}$$

Finally, significance of mediation was tested by obtaining standard errors of direct and indirect effects of objective social class on private-sphere green behavior using 5000 bootstrap replications. We used a mediation analysis model (Model 4 in Hayes’s PROCESS macro) to assess the total, direct and indirect impacts of objective social class on private-sphere green behavior [76].

## 4. Results

Table 1 shows a descriptive overview of the variables considered in this study. The average score in private-sphere green behavior is 11.86 (range from 3 to 15), indicating a relatively high green behavior frequency for Chinese people. As for the social class factors, the mean average score of the respondent’s perceived social status is 4.30, ranging from 1 to 10, indicating that most of the respondents hold a moderate middle subjective class position. The objective social class measured by the logarithm of income had a mean value of 10.32. The result shows that respondents have average and above average concern for the environment; the average score is 28.36, ranging from 8 to 40. About 46 percent of the respondents are male, the average age of the respondents is 51.59, 70 percent of the respondents are in marriage, and a very small sample of the respondents have religious affiliation (6%).

Table 2 displays a correlation matrix between the dependent, independent and control variables. It provides the first evidence that social class plays a significant role in explaining green behavior. Overall, there is bivariate evidence that social class positively correlates with green behavior in private sphere. Positive relationships were found between objective social class, perceived social status and private-sphere green behavior (r = 0.088, *p* < 0.01; r = 0.080, *p* < 0.01, respectively). Environmental concern exhibits high correlations with private-sphere green behavior (r = 0.251, *p* < 0.01). Gender and age correlate with green behavior (r = 0.064, *p* < 0.01; r = −0.089, *p* < 0.01, respectively), among which age negatively correlates with green behavior. However, marriage and religious affiliation are found to have no significant correlations with private-sphere green behavior.

In Table 3, the findings of the OLS models of social class and private-sphere green behavior are presented. As shown in the table, Model 1 is the baseline model that included all control variables. Results revealed that females tend to engage in more green behavior than male respondents (β = 0.330, *p* < 0.01). Individuals who are in marriage have more private-sphere green behavior practices (β = 0.255, *p* < 0.05). The result is partly consistent with prior studies, that showed married individuals tend to protect the environment because of their care about future environmental quality for the next generations. Age has significantly negative effect on the dependent variables. As age increases, the willingness to conduct green behavior decreases (β = −0.013, *p* < 0.01). There is no evidence that religious affiliation affects the practice of green behavior.

We then entered the focal predictors of social class and repeated the procedure for each other indicator, as shown in models 2–6. Among the social class indicators, objective social class has a significantly positive effect on private-sphere green behavior; a unit increase in objective social class leads to an average 0.116 unit increase in private-sphere green behavior (*p* < 0.01), Hypothesis 1 is supported by this result. Subjective social class, as we measured using the perceived social status, does significantly influence private-sphere green behavior (β = 0.103, *p* < 0.01). Thus, we empirically testify Hypothesis 2. Environmental concern is another important factor on private-sphere green behavior. A unit increase in environmental concern leads to an average 0.116 unit increase in private-sphere green behavior (*p* < 0.01). This result supports Hypothesis 4. In Model 5, we incorporate both objective social class and perceived social status into the model and the effect of objective social class shows a downturn from 0.116 to 0.094, with a significance of 0.05. In addition, there is a significant effect of perceived social status in Model 5, with the effect size of 0.095 (*p* < 0.01). In Model 6, we incorporated both objective social class and environmental concern into the model. The results show that the effect of objective social class is no longer significant, with environmental concern significantly influencing private-sphere green behavior (β = 0.114, *p* < 0.01).

Table 3 shows the effect of objective social class, perceived social status and environmental concern on green behavior. It does not reveal the means by which objective social class influences green behavior through perceived social status and environmental concern. Thus, we examined the mediating effect of perceived social status and environmental concern using stepwise regression models. Table 4 displays the results of mediating effects using stepwise regression models. The table shows that the effects of objective class on private-sphere green behavior is partially mediated by perceived social status (*c* = 0.116, *p* = 0.000; *c*’ = 0.094, *p* = 0.016); this result supports Hypothesis 3. In the examination of environmental concern, the results show that environmental concern serves as a full mediator between objective social class and private-sphere green behavior (*c* = 0.116, *p* = 0.000; *c*’= 0.050, *p* = 0.193), thus, Hypothesis 5 is supported.

The process to examine mediating effect using stepwise regression has its deficiencies. First, in the method of stepwise regression, the *p* value is estimated by three separated regression models. However, the estimation of each regression model is conditional on different combinations of predictors. With this in mind, we further examined the mediating effect of perceived social status and environmental concern using the Bootstrap method proposed by Preacher and Hayes [76]. Table 5 shows that the direct effect of objective social class (*c*’ pathway) on the dependent variable (i.e., private-sphere green behavior) and the total effect of objective social class (*c* pathway) was mediated via perceived social status (a1b1 indirect pathway) and environmental concern (a2b2 pathway). As a result, objective social class is linked to both perceived social status (a1 pathway) (β = 0.226, *p* < 0.01) and environmental concern (β = 0.576, *p* < 0.01). Furthermore, perceived social status is favorably related to private-sphere green behavior (b1 pathway) (β = 0.092, *p* < 0.01), and environmental concern is positively related to private-sphere green behavior (b2 pathway) (β = 0.114, *p* < 0.01). Because the 95% confidence interval (CI) does not include zero, bootstrapping analysis reveals the substantial indirect influence of objective social class on private-sphere green behavior through perceived social status (a1b1 = 0.208 (95% CI; LLCI = 0.009, ULCI = 0.034)) and environmental concern (a2b2 = 0.066 (95% CI; LLCI = 0.046, ULCI = 0.088). The total indirect effect was 0.086 (95% CI; LLCI = 0.062, ULCI = 0.113), with environmental concern contributing more than perceived social status. The model revealed that there is no direct effect (*c*’ pathway) on private-sphere green behavior (β = 0.029, *p* = 0.448), implying that the cumulative mediating effect of perceived social status and environmental concern is full (full mediation). Figure 2 shows our path model diagram with the results of parallel mediation analysis.

## 5. Discussion

Existing studies show only limited effects of social class on private-sphere green behavior. Amid ongoing debate about who is most concerned about the environment, examining social class and green behavior is necessary. To this end, our paper sought to address two related research questions in China. First, is there a relationship between social class and private-sphere green behavior? In this light, we empirically testify the effect of different types of social class, both objective and subjective, on private-sphere green behavior. The results of this study support the notion that higher class individuals tend to engage more in private-sphere green behavior than lower class individuals. Similarly, people with perceived higher social status significantly promote green behavior in the private sphere, providing evidence to limited prior studies [77].

Second, what are the mediating mechanisms underlying the relationship between social class and private-sphere green behavior? Our study finds that there is a positive relationship between objective social class and private-sphere green behavior, and that perceived social status and environmental concern fully mediate the relationship between objective social class and private-sphere green behavior. These findings are found to be consistent with prior studies [78]. For instance, Ott and Soretz (2018) found that wealth is a significant predictor of an individual’s attitude toward the environment and that wealthier individuals tend to conduct more green behaviors [8]. Li and Chen (2018) have investigated the effect of income on environmental concerns. As social class represents the amounts of social resources that individual take up in a society, higher class people have more capacity to afford the extra burden of being green [79]. What is more, higher perceptions of status elicit green behaviors in order to demonstrate wealth and identity. This, in turn, encourages individuals’ environmental consciousness to protect the environment [58].

## 6. Conclusions

We summarize the empirical findings and arrive at the following key conclusions: (1) Objective social class has a significant and positive effect on private-sphere green behavior, indicating that higher class individuals tend to engage in more green behavior than lower class individuals. (2) Perceived status rank has a positive effect on private-sphere green behavior. Individuals with perceived higher class in society tend to favor more eco-friendly behaviors in their daily lives. (3) Perceived social status mediates the relationship between objective social class and green behavior in the private sphere; people who tend to have a perceived higher status rank in the society are inclined to be greener. (4) Environmental concern significantly influences private-sphere green behavior and is also mediated by the relationship between objective social class and private-sphere green behavior. This indicate that the influence of objective social class on green behavior is due to enhanced environmental concern.

## 7. Limitations and Future Research

The paper does, however, have several shortcomings. The first is that we did not use moderated models to examine variations of perceived status due to the limitations of the data. Moderating the analysis between objective social class and perceived social status would lead to a better understanding about how interactions of objective and subjective social class influence green behavior. Thus, interactions explaining how social class impacts green behaviors requires further analysis in future studies. Second, though we did find social class positively influence green behavior in the private sphere, we did not discuss the influence of social class on green behavior in the public sphere due to the ambiguous definitions about public-sphere green behavior. We wish to discuss the influence of social class on green behavior in both spheres in future research. The final shortcoming is that we only established simplified associations between social class and green behaviors rather than a causal relationship. This might be enhanced by modifying the regression models into a more sophisticated statistical models and accounting for more comprehensive issues.

## Figures and Tables

**Figure 1 ijerph-20-04329-f001:**
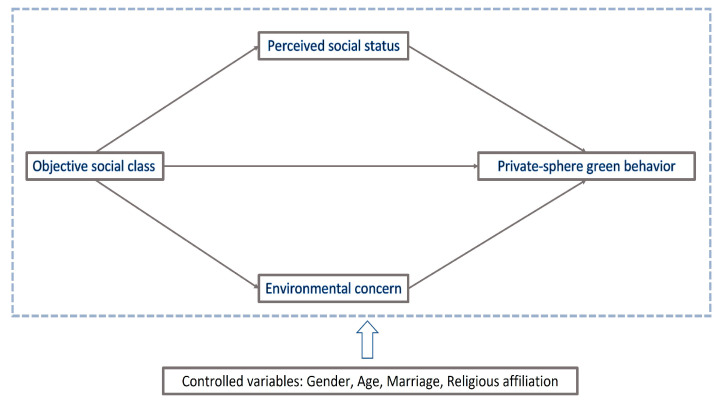
Conceptual framework of social class and private-sphere green behavior.

**Figure 2 ijerph-20-04329-f002:**
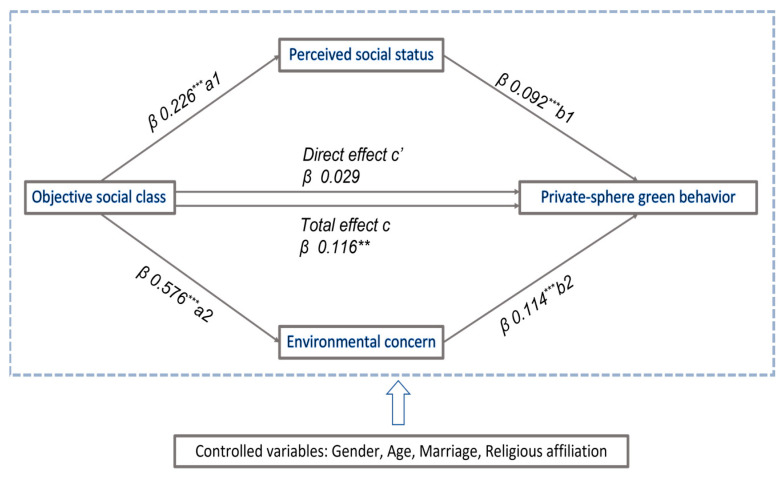
Path model diagram with the results of parallel mediation analysis. Notes: * *p* < 0.1, ** *p* < 0.05, *** *p* < 0.01.

**Table 1 ijerph-20-04329-t001:** Descriptive statistics of all studied variables.

Variable (N = 2739)	Mean	SD	Min	Max
private-sphere green behavior	11.86	2.36	3	15
perceived social status	4.30	1.83	1	10
objective social class	10.32	1.22	4.62	16.12
environmental concern	28.36	4.88	8	40
gender	0.46	0.50	0	1
marriage	0.70	0.46	0	1
age	51.59	17.62	18	94
religious	0.06	0.25	0	1

**Table 2 ijerph-20-04329-t002:** Correlations matrix.

	(1)	(2)	(3)	(4)	(5)	(6)	(7)	(8)
(1) private-sphere green behavior	1							
(2) objective social class	0.088 ***	1						
(3) perceived social status	0.080 ***	0.136 ***	1					
(4) environmental concern	0.251 ***	0.216 ***	0.028	1				
(5) gender	0.064 ***	0.057 ***	−0.024	0.065 ***	1			
(6) age	−0.086 ***	−0.299 ***	−0.005	−0.266 ***	0.044 **	1		
(7) marriage	0.026	0.001	0.021	−0.046 **	−0.025	0.216 ***	1	
(8) religious	−0.009	−0.022	0.022	−0.020	−0.045 **	0.013	−0.011	1

Notes: * *p* < 0.1, ** *p* < 0.05, *** *p* < 0.01.

**Table 3 ijerph-20-04329-t003:** Ordinary Least Square regression models private-sphere green behavior.

Variables	Model 1	Model 2	Model 3	Model 4	Model 5	Model 6
gender	0.330 ***	0.310 ***	0.339 ***	0.244 ***	0.322 ***	0.236 ***
	0.091	0.091	0.090	0.088	0.091	0.089
age	−0.013 ***	−0.011 ***	−0.013 ***	−0.005 *	−0.011 ***	−0.004
	0.003	0.003	0.003	0.003	0.003	0.003
marriage	0.255 **	0.234 **	0.246 **	0.237 **	0.229 **	0.228 **
	0.101	0.101	0.101	0.098	0.101	0.098
religious	−0.040	−0.032	−0.056	−0.009	−0.048	−0.006
	0.184	0.184	0.184	0.179	0.184	0.179
objective social class		0.116 ***			0.094 **	0.050
		0.039			0.039	0.038
perceived social status			0.103 ***		0.095 ***	
			0.025		0.025	
environmental concern				0.116 ***		0.114 ***
				0.009		0.009
constant	12.223 ***	10.923 ***	11.779 ***	8.529 ***	10.755 ***	8.025 ***
	0.151	0.462	0.184	0.331	0.463	0.510
N	2703	2703	2703	2703	2703	2703
Adjusted-R2	0.013	0.016	0.019	0.066	0.021	0.066

Notes: * *p* < 0.1, ** *p* < 0.05, *** *p* < 0.01.

**Table 4 ijerph-20-04329-t004:** Results of mediating effects by stepwise regression models.

M1: Perceived Social Status	Private-Sphere Green Behavior		
	Beta Regression Equation	SE	*p*-Value
Step 1:	Y = 0.116X + e1	0.037	0.003
Step 2:	M1 = 0.226X + e2	0.030	0.000
Step 3:	Y = 0.095M1 + 0.094X + e3	0.025	0.016
		0.039	0.000
M2: environmental concern	Beta regression equation	SE	*p*-value
Step 1:	Y = 0.116X + e1	0.037	0.003
Step 2:	M2 = 0.576X + e2	0.077	0.000
Step 3:	Y = 0.114M1 + 0.050X + e3	0.009	0.000
		0.038	0.193

**Table 5 ijerph-20-04329-t005:** Results of the parallel mediating analysis.

	Private-Sphere Green Behavior		(Bootstrap = 5000)		
	Coeff.	*SE*	*p*-Value	95% CI	
				Lower	Upper
Objective social class (OSC)					
Total effect (*c* pathway)	0.116	0.039	0.030	0.039	0.192
H3: Direct pathway (*c*’ pathway)	0.029	0.039	0.448	−0.046	0.105
OSC to PSS (a1 pathway)	0.226	0.359	0.000	1.068	2.476
OSC to EC (a2 pathway)	0.576	0.920	0.000	23.534	27.140
Total indirect effects	0.086			0.062	0.113
M1: perceived social status (PSS)					
PSS to prigb (b1 pathway)	0.092	0.024	0.000	0.045	0.139
H3: indirect effect of PSS (a1b1 pathway)	0.208			0.009	0.034
M2: Environmental concern (EC)					
EC to prigb (b2 pathway)	0.114	0.024	0.000	0.096	0.133
H5: indirect effect of EC (a2b2 pathway)	0.066			0.046	0.088

Notes: OSC indicates “objective social class”, PSS indicates “perceived social status”; EC indicates “environmental concern”; prigb indicates “private-sphere green behavior” in Table 5.

## Data Availability

All data permissions have been obtained and all data in the paper are copyright free.

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
