# Peer review of "Social Class and Private-Sphere Green Behavior in China: The Mediating Effects of Perceived Status and Environmental Concern"

_ijerph, 2023, doi:10.3390/ijerph20054329_

Round 1

Reviewer 1 Report

The overall quality of the manuscript is high.

There is room for improvement in discussion of this article. It's necessary to summarize and refine the theory based on the findings of this research, and form dialogue with relevant theories.

Author Response

Dear Reviewers:

       On behalf of my co-authors, we thank you very much for giving us an opportunity to revise our manuscript. We appreciate the editor and reviewers very much for the positive and constructive comments and suggestions on our manuscript entitled "Social Class and Green Behavior in China: The Mediating Effects of perceived Status and Environmental Concerns" (ijerph-2237927). In this revised version, we have addressed all reviewers’ questions. A point-by-point response to the reviewer's comments is attached below, and the changes are marked in RED in the manuscript. We hope that these amendments will successfully address you concerns and requirements. We are looking forward to hearing from you soon.

      Sincerely,

      Long Niu

Response to Reviewer 1 Comments

Point 1: There is room for improvement in discussion of this article. It's necessary to summarize and refine the theory based on the findings of this research, and form dialogue with relevant theories.

Response 1:Thank you for your good suggestions. We have reorganized the discussion and conclusion part to enhance our discussion, conclusions, and limitations and future research in the final part of this article, and form dialogue with previous studies.

Reviewer 2 Report

Dear authors,

   Thanks for the opportunity to review this paper. It is a relevant topic and is supposed to provide some valuable insights.  There are some suggestions for improvement.  First, there is no any research model proposed in this study although some regressions were run on investigated independent factors.  I suggest to include the research model and justify it.  Having the model presented in Figure would be helpful.  The  discussion and conclusion are under developed and must be strengthened.  The theoretical and empirical implications are missing.   The limitations and future studies section is missing!   All these need to addressed thoroughly for further review consideration.  

Author Response

Dear Reviewers:

      On behalf of my co-authors, we thank you very much for giving us an opportunity to revise our manuscript. We appreciate the editor and reviewers very much for the positive and constructive comments and suggestions on our manuscript entitled "Social Class and Green Behavior in China: The Mediating Effects of perceived Status and Environmental Concerns" (ijerph-2237927). In this revised version, we have addressed all reviewers’ questions. A point-by-point response to the reviewer's comments is attached below, and the changes are marked in RED in the manuscript. We hope that these amendments will successfully address you concerns and requirements. We are looking forward to hearing from you soon.

     Sincerely,

     Long Niu

Response to Reviewer 2 Comments

Point 1: First, there is no any research model proposed in this study although some regressions were run on investigated independent factors. I suggest to include the research model and justify it. Having the model presented in Figure would be helpful.

Response 1:Thank you for you good suggestions. We have added a research model (see Fig. 1) and justify the hypotheses one by one. The outcome exhibits in the Fig. 2.

Point 2: The discussion and conclusion are under developed and must be strengthened.  The theoretical and empirical implications are missing. The limitations and future studies section is missing!  

Response 2: Thank you for you good suggestions. We have reorganized the discussion and conclusion part to enhance our discussion, conclusions, and limitations and future research in the final part of this article.

Reviewer 3 Report

This paper investigated the correlations between social class and green behaviors with ordinary least-square regression model. I think this paper is an interesting and hot topic. But it needs a major revision before accepted. The following changes are recommended to help improve things:

1. In the process of research design, the author put forward 10 hypotheses, which were too many and did not highlight the focus of the paper. In addition, hypotheses 1 and 2 are correlations, not causation, and there is no need to make hypotheses; Hypotheses 6, 8, and 10, which are also presented in empirical results, have not been verified. Based on these reasons, the authors should reorganize the research hypotheses, put forward the core research hypotheses of the paper, add a figure of theoretical mechanism analysis to illustrate the relationship among all the hypotheses and then demonstrate them one by one. In addition, there are not robustness tests of the basic empirical results.

2. In the part of Method, since the authors did not present the model, this is a major problem. Without a model, how can there be evidence?

3. In the process of mechanism testing, the paper does not indicate whether control variables are added, and control variables need to be added in this part.

4.In table 2, there is no parentheses, but in tote with parentheses. Also, in general, we would say p< 0.05, p < 0.01, p < 0.1,but author said p< 0.001,I don’t know why. In table 5, Why didn't the author report the standard error and P value, but changed it to T value, which led to inconsistency in the full textï¼›std don’t need to add parentheses, because std is separate column,

5. Multiple citations of the same reference are not a good choice. It is recommended to add the latest references.

6. Please check the full text to further improve the language expression. For example, the literature review could use less statements like "who argues" or "who finds", making a comprehensive description is more systematic; Line 389, “models 2-5 and models 7-10”, numbers without parentheses; Line 437-442, where was third? 

Author Response

Dear Reviewers:

      On behalf of my co-authors, we thank you very much for giving us an opportunity to revise our manuscript. We appreciate the editor and reviewers very much for the positive and constructive comments and suggestions on our manuscript entitled "Social Class and Green Behavior in China: The Mediating Effects of perceived Status and Environmental Concerns" (ijerph-2237927). In this revised version, we have addressed all reviewers’ questions. A point-by-point response to the reviewer's comments is attached below, and the changes are marked in RED in the manuscript. We hope that these amendments will successfully address you concerns and requirements. We are looking forward to hearing from you soon.

     Sincerely,

     Long Niu

Response to Reviewer 3 Comments

Point 1: In the process of research design, the author put forward 10 hypotheses, which were too many and did not highlight the focus of the paper. In addition, hypotheses 1 and 2 are correlations, not causation, and there is no need to make hypotheses; Hypotheses 6, 8, and 10, which are also presented in empirical results, have not been verified. Based on these reasons, the authors should reorganize the research hypotheses, put forward the core research hypotheses of the paper, add a figure of theoretical mechanism analysis to illustrate the relationship among all the hypotheses and then demonstrate them one by one. In addition, there are not robustness tests of the basic empirical results.

Response 1: Thanks for the Reviewer’s kind suggestion. We do recognize that there are too many hypotheses in this paper, and since there is no evidence showing that social class directly correlates with public-sphere green behavior, so we focus our dependent variable of private-sphere green behavior rather than both spheres to form our core research hypotheses of the paper as follows:

Hypothesis 1: There is a positive relationship between objective social class and private-sphere green behavior.

Hypothesis 2: There is a positive relationship between perceived status and private-sphere green behavior.

Hypothesis 3: Perceived status mediates the effect between objective social class and private-sphere green behavior.

Hypothesis 4: There is positive relationship between environmental concern and private-sphere green behavior.

Hypothesis 5: Environmental concern mediates the relationship between objective social class and private-sphere green behavior.

In addition, we have changed the statement about “causations” into “correlations” or “relationship” in the statement of hypotheses.

Besides, we have added our conceptual framework (see Fig. 1) in this paper to make our statement and research pathway much clearer and more straightforward.

Point 2: In the part of Method, since the authors did not present the model, this is a major problem. Without a model, how can there be evidence?

Response 2: Thank you for your kind reminders. In Part 3.3, Line 342-Line 373, we have reorganized the description of our analytical strategy, and add OLS model and stepwise regression models as our analytical strategy to make it evidenced.

Point 3: In the process of mechanism testing, the paper does not indicate whether control variables are added, and control variables need to be added in this part.

Response 3: Thank you for your kind reminders. We have added a conceptual framework (see Fig. 1) of this paper, and Fig. 1 depicts that all the process of analytical strategy is under the control of controlled variables.

Point 4: In table 2, there is no parentheses, but in tote with parentheses. Also, in general, we would say p< 0.05, p < 0.01, p < 0.1,but author said p< 0.001,I don’t know why. In table 5, Why didn't the author report the standard error and P value, but changed it to T value, which led to inconsistency in the full textï¼›std don’t need to add parentheses, because std is separate column,

Response 4:Thanks for the reviewer’s kind reminds. We have substituted Table 2 with a new one, and it’s our carelessness to say p<0.001, we have modified all the significance level to * p < 0.1, ** p < 0.05, *** p < 0.01 in Line 399 and Line 428. Besides, in Table 4, we have changed the T-value into P-value to be consistent with the full text and avoid the confusion. Finally, we erased the parentheses of the Std. in Table 3.

Point 5: Multiple citations of the same reference are not a good choice. It is recommended to add the latest references.

Response 5:Thanks for the reviewer’s kind suggestions. We have substituted some multiple citations with some latest references.

Point 6: Please check the full text to further improve the language expression. For example, the literature review could use less statements like "who argues" or "who finds", making a comprehensive description is more systematic; Line 389, “models 2-5 and models 7-10”, numbers without parentheses; Line 437-442, where was third? 

Response 6:Thank you for your kind reminders. This paper has been professionally proofread for language, and we changed the statements like “who argues” or “who finds” to “indicates “or “indicating that” to make comprehensive description more systematic. Besides, we have made a major revision to the regression models and reorganized the description of the OLS regression part, see Part 4 Results.

Reviewer 4 Report

Social class and green behaviors in China: the mediating effects of perceived status and environmental concern

This study investigates the mediating effect of environmental concern between social class and green behavior due to the breadth of environmental problems, and the result show that the influence of social class on green behaviors not only mediated by perceived social status but also mediated by individual’s environmental concern. There are some points for your references and considers in your revise version

1.          This is an interesting piece of “Social class and green behaviors in China” work. Please underscore the scientific value added/contributions of your paper in your abstract and introduction and address your debate shortly in the abstract. However the contribution should based on the gaps from the prior studies.

2.          The performance literature review is extensive and covers the relevant material with a considerable degree of insight. The paper is very well structured. The material is well presented. I would suggest the author to discuss these references in your context and references. For instance, Ming-Lang Tseng, Shu-Xian Li, Chun-Wei Remen Lin & Anthony SF Chiu (2023) Validating green building social sustainability indicators in China using the fuzzy delphi method, Journal of Industrial and Production Engineering, 40:1, 35-53; and Chunyan Zhu, Xu Guo & Shaohui Zou (2022) Impact of information and communications technology alignment on supply chain performance in the Industry 4.0 era: mediation effect of supply chain integration, Journal of Industrial and Production Engineering, 39:7, 505-520

3.          I would suggest the discussion and result separate into 2 sections

4.          Please also show your conceptual framework and final output in Fiures.

5.          Your conclusions' section needs to underscore the scientific value added of your paper, and/or the applicability of your findings/results, as indicated previously. Basically, you should enhance your findings, limitations, underscore the scientific value added of your paper, and/or the applicability of your contributions/shortages and future study in this session.

6.          “marry” should be “marriage”. Please place the questionnaire in your appendix.

Author Response

Dear Reviewers:

      On behalf of my co-authors, we thank you very much for giving us an opportunity to revise our manuscript. We appreciate the editor and reviewers very much for the positive and constructive comments and suggestions on our manuscript entitled "Social Class and Green Behavior in China: The Mediating Effects of perceived Status and Environmental Concerns" (ijerph-2237927). In this revised version, we have addressed all reviewers’ questions. A point-by-point response to the reviewer's comments is attached below, and the changes are marked in RED in the manuscript. We hope that these amendments will successfully address you concerns and requirements. We are looking forward to hearing from you soon.

     Sincerely,

     Long Niu

Response to Reviewer 4 Comments

Point 1: This is an interesting piece of “Social class and green behaviors in China” work. Please underscore the scientific value added/contributions of your paper in your abstract and introduction and address your debate shortly in the abstract. However the contribution should based on the gaps from the prior studies.

Response 1: Thank you for your constructing advice. We underscored the scientific value and contributions in the abstract by adding “The present research provides the insights about how social class and its psychological manifestations correlate with private-green behavior in China for the first time, and our results suggest that more social context factors should be taken into account when identifying the factors of promoting pro-environmental behavior in China.” in the ABSTRACT part, Line 20-23. And in the INTRODUCTION part, we added, ” The present study provides a nuanced look into how different social class, both objective and subjective, influences green behaviors in China. Furthermore, by analyzing the mediating effect of perceived status between objective social class and green behaviors, this study takes a closer look of the psychological manifestation of objective social class in influencing individual’s green behaviors.” to underscore of our research and values, see Line 98-103.

Point 2: The performance literature review is extensive and covers the relevant material with a considerable degree of insight. The paper is very well structured. The material is well presented. I would suggest the author to discuss these references in your context and references. For instance, Ming-Lang Tseng, Shu-Xian Li, Chun-Wei Remen Lin & Anthony SF Chiu (2023) Validating green building social sustainability indicators in China using the fuzzy delphi method, Journal of Industrial and Production Engineering, 40:1, 35-53; and Chunyan Zhu, Xu Guo & Shaohui Zou (2022) Impact of information and communications technology alignment on supply chain performance in the Industry 4.0 era: mediation effect of supply chain integration, Journal of Industrial and Production Engineering, 39:7, 505-520

Response 2: Thanks for the suggestions. You have enlightened us that green building is an important indicator that construal green behaviors of Chinese people. We have cited this article in our introduction part in Line 39-40. Besides, the second paper that you recommended inspired us how to perform mediating analysis in the article, we’ve learned a lot and improve our method in the mediating analysis.

Point 3: I would suggest the discussion and result separate into 2 sections

Response 3: Thank you for your kind suggestions. Inspired by this advice, we have separated original discussion part into DISCUSSION, CONCLUSIONS, and LIMITATIONS AND FUTURE REARCH in the final of our article

Point 4:  Please also show your conceptual framework and final output in Fiures.

Response 4: Thanks for the suggestion. We have added our conceptual framework in this paper (see Fig. 1) to make our statement and research pathway much clearer and more straightforward.

Point 5:  Your conclusions' section needs to underscore the scientific value added of your paper, and/or the applicability of your findings/results, as indicated previously. Basically, you should enhance your findings, limitations, underscore the scientific value added of your paper, and/or the applicability of your contributions/shortages and future study in this session.

Response 5: Thank you for your kind suggestions. Acknowledged that our conclusions’ section underscores the scientific value, we have rewritten this part and separated it into three parts by DISCUSSION, CONCLUSIONS, and LIMITATIONS AND FUTURE RESEARC in the final of the article to enhance the findings, limitations and scientific value of this paper.

Point 6:  “marry” should be “marriage”. Please place the questionnaire in your appendix.

Response 6: Thanks for the Reviewer’s suggestion. We’ve changed “marry” to “marriage” to correspond with variable definition. Besides, we also place our questionnaire in the appendix.

Round 2

Reviewer 2 Report

   After reviewing the revised manuscript, I feel the authors have tried to revise the paper to address the concerns raised by the reviewers.  The quality has been improved to be considered to accept this submission. 

Reviewer 3 Report

The authors have made a comprehensive revision according to the review comments. I have no problem, I suggest receiving it. In addition, please pay attention to the details again, for example, Figures 1 and 2 are not clear.